# DEEP $k$-NN LABEL SMOOTHING IMPROVES STABILITY OF NEURAL NETWORK PREDICTIONS

## ABSTRACT

Training modern neural networks is an inherently noisy process that can lead to high *prediction churn*– disagreements between re-trainings of the same model due to factors such as randomization in the parameter initialization and mini-batches– even when the trained models all attain high accuracies. Such prediction churn can be very undesirable in practice. In this paper, we present several baselines for reducing churn and show that utilizing the $k$-NN predictions to smooth the labels results in a new and principled method that often outperforms the baselines on churn while improving accuracy on a variety of benchmark classification tasks and model architectures.

## 1 INTRODUCTION

Deep neural networks (DNNs) have proved to be immensely successful at solving complex classification tasks across a range of problems. Much of the effort has been spent towards improving their predictive performance (i.e. accuracy), while comparatively little has been done towards improving the *stability* of training these models. Modern DNN training is inherently noisy due to factors such as the random initialization of network parameters, the mini-batch ordering, and effects of various data augmentation or pre-processing tricks, all of which are exacerbated by the non-convexity of the loss surface. This results in local optima corresponding to models that have very different predictions on the same data points. This may seem counter-intuitive, but even when the different runs all produce very high accuracies for the classification task, their predictions can still differ quite drastically as we will show later in the experiments. Thus, even an optimized training procedure can lead to high *prediction churn*, which refers to the proportion of sample-level disagreements between classifiers caused by different runs of the same training procedure[1].

In practice, reducing such predictive churn can be critical. For example, in a production system, models are often continuously improved on by being trained or retrained with new data or better model architectures and training procedures. In such scenarios, a candidate model for release must be compared to the current model serving in production. Oftentimes, this decision is conditioned on more than just overall offline test accuracy– in fact, oftentimes the offline metrics are not completely aligned with actual goal, especially if these models are used as part of a larger system (e.g. maximizing offline click-through rate vs. maximizing revenue or user satisfaction). As a result, these comparisons oftentimes require extensive and costly live experiments, requiring human evaluation in situations where the candidate and the production model disagree (i.e. in many situations, the true labels are not available without a manual labeler). In these cases, it can be highly desirable to lower prediction churn.

Despite the practical relevance of lowering predictive churn, there has been surprisingly little work done in this area, which we highlight in the related work section. In this work, we focus on predictive churn reduction under retraining the same model architecture on an identical train and test set. Our main contributions are as follows:

- We provide one of the first comprehensive analyses of baselines to lower prediction churn, showing that popular approaches designed for other goals are effective baselines for churn reduction, even compared to methods designed for this goal.

---

[1]Concretely, given two classifiers applied to the same test samples, the prediction churn between them is the fraction of test samples with different predicted labels.

- We improve label smoothing, a *global* smoothing method popular for improving model confidence scores, by utilizing the *local* information leveraged by the $k$-NN labels thus introducing $k$-NN label smoothing which we show to often outperform the baselines on a wide range of benchmark datasets and model architectures.

- We show new theoretical results for the $k$-NN labels suggesting the usefulness of the $k$-NN label. We show under mild nonparametric assumptions that for a wide range of $k$, the $k$-NN labels uniformly approximates the Bayes-optimal label and when $k$ is tuned optimally, achieves the minimax optimal rate. We also show that when $k$ is linear in $n$, the distribution implied by the $k$-NN label approximates the original distribution smoothed with an *adaptive* kernel.

## 2 RELATED WORKS

Our work spans multiple sub-areas of machine learning. The main problem this paper tackles is reducing prediction churn. In the process, we show that label smoothing is an effective baseline and we improve upon it in a principled manner using deep $k$-NN label smoothing.

**Prediction Churn.** There are only a few works which explicitly address prediction churn. Fard et al. (2016) proposed training a model so that it has small prediction instability with future versions of the model by modifying the data that the future versions are trained on. They furthermore propose turning the classification problem into a regression towards corrected predictions of an older model as well as regularizing the new model towards the older model using example weights. Cotter et al. (2019); Goh et al. (2016) use constrained optimization to directly lower prediction churn across model versions. Simultaneously training multiple identical models (apart from initialization) while tethering their predictions together via regularization has been proposed in the context of distillation (Anil et al., 2018; Zhang et al., 2018; Zhu et al., 2018; Song & Chai, 2018) and robustness to label noise (Malach & Shalev-Shwartz, 2017; Han et al., 2018). This family of methods was termed "co-distillation" by Anil et al. (2018), who also noted that it can be used to reduce churn in addition to improving accuracy. In this paper, we show much more extensively that co-distillation is indeed a reasonable baseline for churn reduction.

**Label smoothing.** Label smoothing (Szegedy et al., 2016) is a simple technique that proposes to train a model the model on the soft labels obtained by a convex combination of the hard true label and the soft uniform distribution across all the labels. It has been shown that it prevents the network from being over-confident and leads to better confidence calibration (Müller et al., 2019). Here we show that label smoothing is a reasonable baseline for reducing prediction churn, and we moreover enhance it for this task by smoothing the labels *locally* via $k$-NN rather than a the pure *global* approach mixing with the uniform distribution.

$k$-**NN Theory.** The theory of $k$-NN classification has a long history (e.g. Fix & Hodges Jr (1951); Cover (1968); Stone (1977); Devroye et al. (1994); Chaudhuri & Dasgupta (2014)). To our knowledge, the most relevant $k$-NN classification result is by Chaudhuri & Dasgupta (2014), who show statistical risk bounds under similar assumptions as used in our work. Our analysis shows finite-sample $L_\infty$ bounds on the $k$-NN labels, which is a stronger notion of consistency as it provides a uniform guarantee, rather than an *average* guarantee as is shown in previous works under standard risk measures such as $L_2$ error. We do this by leveraging recent techniques developed in Jiang (2019) for $k$-NN regression, which assumes an additive noise model instead of classification. Moreover, we provide to our knowledge the first consistency guarantee for the case where $k$ grows linearly with $n$.

**Deep $k$-NN.** $k$-NN is a classical method in machine learning which has recently been shown to be useful when applied to the intermediate embeddings of a deep neural network (Papernot & McDaniel, 2018) to obtain more calibrated and adversarially robust networks. This is because standard distance measures are often better behaved in these representations leading to better performance of $k$-NN on these embeddings than on the raw inputs. Jiang et al. (2018) uses nearest neighbors on the intermediate representations to obtain better uncertainty scores than softmax probabilities and Bahri et al. (2020) uses the $k$-NN label disagreement to filter noisy labels for better training. Like these works, we also leverage $k$-NN on the intermediate representations but we show that utilizing the $k$-NN labels leads to lower prediction churn.

## 3 ALGORITHM

Suppose that the task is multi-class classification with $L$ classes and the training datapoints are $(x_1, y_1), ..., (x_n, y_n)$, where $x_i \in \mathcal{X}$, and $\mathcal{X}$ is a compact subset of $\mathbb{R}^D$ and $y_i \in \mathbb{R}^L$, where represents the one-hot vector encoding of the label– that is, if the $i$-th example has label $j$, then $y_i$ has $1$ in the $j$-th entry and $0$ everywhere else. Then we give the formal definition of the smoothed labels:

**Definition 1** (Label Smoothing). *Given label smoothing parameter $0 \le a \le 1$, then the smoothed label $y$ is (where $\mathbf{1}_L$ denotes the vector of all $1$'s in $\mathbb{R}^L$).*

$$y_a^{LS} := (1 - a) \cdot y + \frac{a}{L} \cdot \mathbf{1}_L.$$

We next formally define the $k$-NN label, which is the average label of the example's $k$-nearest neighbors in the training set. Let us use shorthand $X := \{x_1, ..., x_n\}$ and $y_i \in \mathbb{R}^L$.

**Definition 2** ($k$-NN label). *Let the $k$-NN radius of $x \in \mathcal{X}$ be $r_k(x) := \inf\{r : |B(x, r) \cap X| \ge k\}$ where $B(x, r) := \{x' \in \mathcal{X} : |x - x'| \le r\}$ and the $k$-NN set of $x \in \mathcal{X}$ be $N_k(x) := B(x, r_k(x)) \cap X$. Then for all $x \in \mathcal{X}$, the $k$-NN label is defined as*

$$\eta_k(x) := \frac{1}{|N_k(x)|} \sum_{i=1}^{n} y_i \cdot \mathbf{1}\left[x_i \in N_k(x)\right].$$

The label smoothing method can be seen as performing a global smoothing. That is, every label is equally transformed towards the uniform distribution over all labels. While it seems almost deceptively simple, it has only recently been shown to be effective in practice, specifically for better calibrated networks. However, since this smoothing technique is applied equally to all datapoints, it fails to incorporate local information about the datapoint. To this end, we propose using the $k$-NN label, which smooths the label across its nearest neighbors. We show theoretically that the $k$-NN label can be a strong proxy for the Bayes-optimal label, that is, the best possible prediction one can make given the uncertainty. In other words, compared to the true label (or even the label smoothing), the $k$-NN label is robust to variability in the data distribution and provides a more stable estimate of the label than the original hard label which may be noisy. Training on such noisy labels have been shown to hurt model performance (Bahri et al., 2020) and using the smoothed labels can help mitigate these effects. To this end, we define $k$-NN label smoothing as follows:

**Definition 3** ($k$-NN label smoothing). *Let $0 \le a, b \le 1$ be $k$-NN label smoothing parameters. Then the $k$-NN smoothed label of datapoint $(x, y)$ is defined as:*

$$y_{a,b}^{kNN} = (1 - a) \cdot y + a \cdot \left(b \cdot \frac{1}{L} \cdot \mathbf{1}_L + (1 - b) \cdot \eta_k(x)\right).$$

We see that $a$ is used to weight between using the true labels vs. using smoothing, and $b$ is used to weight between the global vs. local smoothing. Algorithm 1 shows how $k$-NN label smoothing is applied to deep learning models. Like Bahri et al. (2020), we perform $k$-NN on the network's logits layer.

---

**Algorithm 1** Deep $k$-NN label smoothing

---

**Inputs:** $0 \le a, b \le 1$, Training data $(x_1, y_1), ..., (x_n, y_n)$, model training procedure $\mathcal{M}$.
Train model $M_0$ on $(x_1, y_1), ..., (x_n, y_n)$ with $\mathcal{M}$.
Let $z_1, ..., z_n \in \mathbb{R}^L$ be the logits of $x_1, ..., x_n$, respectively, w.r.t. $M_0$
Let $\widetilde{y_i}$ be the $k$-NN smoothed label of $(z_i, y_i)$ computed w.r.t. dataset $(z_1, y_1), ..., (z_n, y_n)$.
Train model $M$ on $(x_1, \widetilde{y_1}), ..., (x_n, \widetilde{y_n})$ with $\mathcal{M}$.

---

## 4 THEORETICAL ANALYSIS

In this section, we provide theoretical justification for why the $k$-NN labels may be useful. In particular, we show results for two settings, where $n$ is the number of datapoints.

- When $k \ll n$, we show that with appropriate setting of $k$, the $k$-NN smoothed labels approximates the predictions of Bayes-optimal classifier at a minimax-optimal rate.

- When $k = O(n)$, we show that the distribution implied by the $k$-NN smoothed labels is equivalent to the original distribution convolved with an *adaptive* smoothing kernel.

Our results may also reveal insights into why distillation methods (the procedure of training a model on another model's predictions instead of the true labels) can work. Another way of considering the result is that the $k$-NN smoothed label is equivalent to the soft prediction of the $k$-NN classifier. Thus, if one were to train on the $k$-NN labels, it would be essentially distillation on the $k$-NN classifier and our theoretical results show that the labels implied by $k$-NN approximate the predictions of the *optimal* classifier (in the $k \ll n$ setting). Learning the optimal classifier may indeed be a better goal than learning from the true labels, because the latter may lead to overfitting to the sampling noise rather than just the true signal implied by the optimal classifer. While distillation is not the topic of this work, our results in this section may be of independent interest to that area.

For the analysis, we assume the binary classification setting, but it is understood that our results can be straightforwardly generalized to the multi-class setting. The feature vectors are defined on compact support $\mathcal{X} \subseteq \mathbb{R}^D$ and datapoints are drawn as follows: the features vector is drawn from density $p_{\mathcal{X}}$ on $\mathcal{X}$ and the labels are drawn according to the label function $\eta : \mathcal{X} \to [0, 1]$, i.e. $\eta(x) = \mathbb{P}(Y = 1|X = x)$.

### 4.1 $k \ll n$

We make a few mild regularity assumptions for our analysis to hold, which are standard in works analyzing non-parametric methods e.g. Singh et al. (2009); Chaudhuri & Dasgupta (2014); Reeve & Kaban (2019); Jiang (2019); Bahri et al. (2020). The first part ensures that the support $\mathcal{X}$ does not become arbitrarily thin anywhere, the second ensures that the density does not vanish anywhere in the support, and the third ensures that the label function $\eta$ is smooth w.r.t. to its input.

**Assumption 1.** *The following three conditions hold:*

- *Support Regularity: There exists $\omega > 0$ and $r_0 > 0$ such that $Vol(\mathcal{X} \cap B(x, r)) \geq \omega \cdot Vol(B(x, r))$ for all $x \in \mathcal{X}$ and $0 < r < r_0$, where $B(x, r) := \{x' \in \mathcal{X} : |x - x'| \leq r\}$.*

- *Non-vanishing density: $p_{X,0} := \inf_{x \in \mathcal{X}} p_X(x) > 0$.*

- *Smoothness of $\eta$: There exists $0 < \alpha \leq 1$ and $C_\alpha > 0$ such that $|\eta(x) - \eta(x')| \leq C_\alpha |x - x'|^\alpha$ for all $x, x' \in \mathcal{X}$.*

We have the following result which provides a *uniform* bound between the smoothed $k$-NN label $\eta_k$ and the Bayes-optimal label $\eta$.

**Theorem 1.** *Let $0 < \delta < 1$ and suppose that Assumption 1 holds and that $k$ satisfies the following:*

$$2^8 \cdot D \log^2(4/\delta) \cdot \log n \leq k \leq \frac{1}{2} \cdot \omega \cdot p_{X,0} \cdot v_D \cdot r_0^D \cdot n,$$

*where $v_D := \frac{\pi^{D/2}}{\Gamma(d/2+1)}$ is the volume of a $D$-dimensional unit ball. Then with probability at least $1 - \delta$, we have*

$$\sup_{x \in \mathcal{X}} |\eta_k(x) - \eta(x)| \leq C_\alpha \left(\frac{2k}{\omega \cdot v_D \cdot n \cdot p_{X,0}}\right)^{\alpha/D} + \sqrt{\frac{2\log(4D/\delta) + 2D\log(n)}{k}}.$$

In other words, there exists constants $C_1, C_2, C$ depending on $\eta$ and $\delta$ such that if $k$ satisfies

$$C_1 \log n \leq k \leq C_2 \cdot n,$$

then with probability at least $1 - \delta$, ignoring logarithmic factors in $n$ and $1/\delta$:

$$\sup_{x \in \mathcal{X}} |\eta_k(x) - \eta(x)| \leq C \cdot \left(\left(\frac{k}{n}\right)^{\alpha/D} + \frac{1}{\sqrt{k}}\right).$$

Choosing $k \approx n^{2\alpha/(2\alpha+D)}$, gives us a bound of $\sup_{x \in \mathcal{X}} |\eta_k(x) - \eta(x)| \leq \widetilde{O}(n^{-1/(2\alpha+D)})$, which is the minimax optimal rate as established by Tsybakov et al. (1997).

Therefore, the advantage of using the smoothed labels $\eta_k(x_1), ..., \eta_k(x_n)$ instead of the true labels $y_1, ..., y_n$, is that the smoothed labels approximate the Bayes-optimal classifier. Moreover, as shown above, with appropriate setting of $k$, the smoothed labels are a minimax-optimal estimator of the true label function $\eta$. Thus, the smoothed labels provide as good of a proxy for $\eta$ as any estimator possibly can.

As suggested earlier, another way of considering this result is that the original labels may contain considerable noise and thus no single label can be guaranteed reliable. Using the smoothed label instead mitigates this effect and allows us to train the model to match the label function $\eta$.

### 4.2 $k$ LINEAR IN $n$

In the previous subsection, we showed the utility of $k$-NN label smoothing as a theoretically sound proxy for the Bayes-optimal labels, which attains statistical consistency guarantees as long as $k$ grows faster than $\log n$ and $k/n \to 0$. Now, we analyze the case where $k$ grows linearly with $n$. In this case, the $k$-NN smoothed labels no longer recover the Bayes-optimal label function $\eta$, but instead an adaptive kernel smoothed version of $\eta$. We make this relationship precise here.

Suppose that $k = \lfloor \beta \cdot n \rfloor$ for some $0 < \beta < 1$. We define the $\beta$-smoothed label function:

**Definition 4** ($\beta$-smoothed label function). *Let $r_\beta(x) := \inf\{r > 0 : \mathcal{P}(B(x,r)) \geq \beta\}$, that is the radii of the smallest ball centered at $x$ with probability mass $\beta$ w.r.t. $P_X$. Then, let $\widetilde{\eta}_\beta(x)$ be the expectation of $\eta$ on $B(x, r_\beta(x))$ w.r.t. $P_X$:*

$$\widetilde{\eta}_\beta(x) := \frac{1}{\beta} \int_{B(x, r_\beta(x))} \eta(x) \cdot P_X(x) dx.$$

We can view $\widetilde{\eta}_\beta$ as an adaptively kernel smoothed version of $\eta$, where adaptivity arises from the density of the point (the more dense, the smaller the bandwidth we smooth it across) and the kernel is based on the density.

We now prove the following result which shows that in this setting $\eta_k$ estimates $\widetilde{\eta}_\beta(x)$. It is worth noting that we need very little assumption on $\eta$ as compared to the previous result because the $\beta$-smoothing of $\eta$ provides a more regular label function; moreover, the rates are fast i.e. $\widetilde{O}(\sqrt{D/n})$.

**Theorem 2.** *Let $0 < \delta < 1$ and $k = \lfloor \beta \cdot n \rfloor$. Then with probability at least $1 - \delta$, we have for $n$ sufficiently large depending on $\beta, \delta$:*

$$\sup_{x \in \mathcal{X}} |\eta_k(x) - \widetilde{\eta}_\beta(x)| \leq 3\sqrt{\frac{2\log(4D/\delta) + 2D\log(n)}{\beta \cdot n}}.$$

## 5 EXPERIMENTS

We now describe the experimental methodology and results for validating our proposed method.

### 5.1 BASELINES

We next detail the suite of baselines we compare against. We tune baseline hyper-parameters extensively, with the precise sweeps and setups available in the Appendix.

- **Control**: Baseline where we train for accuracy without regards to lower churn.
- $\ell_p$ **Regularization**: We control the stability of a model's predictions by simply regularizing them (independently of the ground truth label) using classical $\ell_p$ regularization. The loss function is given by:

$$\mathcal{L}_{\ell_p}(x_i, y_i) = \mathcal{L}(x_i, y_i) + a||f(x_i)||_p^p.$$

  We experiment with both $\ell_1$ and $\ell_2$ regularization.

- **Bi-tempered**: This is a baseline by Amid et al. (2019), originally designed for robustness to label noise. It modifies the standard logistic loss function by introducing two temperature scaling parameters $t_1$ and $t_2$. We apply their "bi-tempered" loss here, suspecting that methods which make model training more robust to noisy labels may also be effective at reducing prediction churn.

- **Anchor**: This is based on a method proposed by Fard et al. (2016) specifically for churn reduction. It uses the predicted probabilities from a preliminary model to smooth the training labels of the second model. We first train a preliminary model $f_{\text{prelim}}$ using regular cross-entropy loss. We then retrain the model using smoothed labels $(1 - a)y_i + af_{\text{prelim}}(x_i)$, thus "anchoring" on a preliminary model's predictions. In our experiments, we train one preliminary model and fix it across the runs for this baseline to reduce prediction churn.

- **Co-distillation**: We use the co-distillation approach presented by Anil et al. (2018), who touched upon its utility for churn reduction. We train two identical models $M_1$ and $M_2$ (but subject to different random initialization) in tandem while penalizing divergence between their predictions. The overall loss is

$$\mathcal{L}_{\text{codistill}}(x_i, y_i) = \mathcal{L}(f_1(x_i), y_i) + \mathcal{L}(f_2(x_i), y_i) + a\Psi(f_1(x_i), f_2(x_i)).$$

In their paper, the authors set $\Psi$ to be cross-entropy:

$$\Psi(p^{(1)}, p^{(2)}) = \sum_{i \in [K]} p_i^{(1)} \log(p_i^{(2)}),$$

but they note KL divergence can be used. We experiment with both cross-entropy and KL divergence. We also tune $w_{\text{codistill}}$, the number of burn-in steps of training before turning on the regularizer.

- **Label Smoothing**: This is the method of Szegedy et al. (2016) defined earlier in the paper. Our proposed method augments global label smoothing by leveraging the local $k$-NN estimates. Naturally, we compare against doing global smoothing only and this serves as a key ablation model to see the added benefits of leveraging the $k$-NN labels.

- **Mixup**: This method proposed by Zhang et al. (2017) generates synthetic training examples on the fly by convex combining random training inputs and their associated labels, where the combination weights are random draws from a Beta$(a, a)$ distribution. Mixup improves generalization, increases robustness to adversarial examples as well as label noise, and also improves model calibration (Thulasidasan et al., 2019).

- **Ensemble**: Ensembling deep neural networks can improve the quality of their uncertainty estimation (Lakshminarayanan et al., 2017; Fort et al., 2019). We consider the simple case where $m$ identical deep neural networks are trained independently on the same training data, and at inference time, their predictions are uniformly averaged together.

## 5.2 DATASETS AND MODELS.

For all datasets, we do not use any data augmentation in order to guarantee that the training data used to across different trainings is held fixed. For all datasets we use the Adam optimizer with default learning rate 0.001. We use a minibatch size of 128 throughout.

- **MNIST**: We train a two-layer MLP with 256 hidden units per layer and ReLU activations for 20 epochs.

- **Fashion MNIST**: We use the same architecture as the one used for MNIST.

- **SVHN**: We use LeNet5 CNN(LeCun et al., 1998) for 30 epochs on the Google Street View Housing Numbers (SVHN) dataset, where each image is cropped to be $32 \times 32$ pixels.

- **CelebA**: CelebA (Liu et al., 2018) is a large-scale face attributes dataset with more than 200k celebrity images, each with 40 attribute annotations. We use the standard train and test splits, which consist of $162770$ and $19962$ images respectively. Images were resized to be $28 \times 28 \times 3$. We select the "smiling" and "high cheekbone" attributes and perform binary classification, training LeNet5 for 20 epochs.

- **Phishing**: To validate our method beyond the image classification setting, we train a two-layer MLP with 256 hidden units per layer on UCI Phishing dataset (Dua & Graff, 2017), which consists of 7406 train and 3649 test examples on a 30-dimensional input feature.

| Dataset | Method | Accuracy % | Churn % | Churn Correct | Churn Incorrect |
|---|---|---|---|---|---|
| SVHN | $k$-NN LS (k=10, a=1, b=0.9) | 88.98 (0.33) | 10.98 (0.28) | 4.64 (0.29) | 62.23 (1.22) |
| | Label Smoothing (a=0.9) | 87.26 (0.73) | 13.46 (0.62) | 5.31 (0.57) | 67.2 (1.44) |
| | Anchor (a=1.0) | 87.17 (0.16) | 12.48 (0.39) | 5.19 (0.2) | 61.66 (1.85) |
| | $\ell_2$ Reg (a=0.5) | 88.16 (0.35) | 11.85 (0.35) | 5.07 (0.16) | 62.73 (2.1) |
| | $\ell_1$ Reg (a=0.2) | 74.18 (3.41) | 22.89 (3.74) | 9.58 (4.04) | 59.36 (5.7) |
| | Co-distill (CE, a=0.5) | 87.64 (0.64) | 12.46 (0.48) | 5.16 (0.51) | 63.82 (1.67) |
| | Co-distill (KL, a=0.5) | 87.52 (0.45) | 13.01 (0.3) | 5.54 (0.33) | 65.44 (1.46) |
| | Bi-tempered ($t_1$=0.5, $t_2$=1) | 88.04 (0.5) | 12.03 (0.3) | 5.26 (0.3) | 62.48 (1.83) |
| | Mixup (a=0.5) | **89.08 (0.18)** | **9.56 (0.16)** | **4.07 (0.15)** | 54.75 (0.95) |
| | Control | 86.64 (0.54) | 14.64 (0.51) | 6.03 (0.5) | 69.59 (1.32) |
| MNIST | $k$-NN LS (k=5, a=0.9, b=0.9) | **98.23 (0.11)** | **1.52 (0.12)** | **0.7 (0.1)** | 47.16 (3.39) |
| | Label Smoothing (a=0.9) | 98.15 (0.07) | 1.65 (0.05) | 0.71 (0.07) | 50.73 (2.62) |
| | Anchor (a=1.0) | 97.72 (0.11) | 2.66 (0.2) | 1.21 (0.14) | 64.51 (4.13) |
| | $\ell_2$ Reg (a=0.5) | 98.08 (0.1) | 1.67 (0.12) | 0.8 (0.08) | 46.65 (3.2) |
| | $\ell_1$ Reg (a=0.01) | 97.67 (0.29) | 2.51 (0.31) | 1.3 (0.27) | 56.8 (2.84) |
| | Co-distill (CE, a=0.2, $n_{warm}$=2k) | 98.08 (0.06) | 2.08 (0.11) | 0.98 (0.07) | 58.6 (3.91) |
| | Co-distill (KL, a=0.05, $n_{warm}$=1k) | 97.98 (0.14) | 2.16 (0.16) | 0.97 (0.13) | 59.56 (3.64) |
| | Bi-tempered ($t_1$=0.9, $t_2$=1.0) | 98.09 (0.2) | 2.04 (0.15) | 1.07 (0.14) | 55.82 (4.32) |
| | Mixup (a=0.2) | 98.17 (0.04) | 1.59 (0.07) | 0.74 (0.04) | 47.8 (2.53) |
| | Control | 97.98 (0.13) | 2.28 (0.13) | 0.96 (0.07) | 63.36 (2.55) |
| Fashion MNIST | $k$-NN LS (k=10, a=1, b=0.5) | 88.89 (0.14) | 6.94 (0.18) | 3.27 (0.15) | 36.26 (1.09) |
| | Label Smoothing (a=0.8) | 88.46 (0.17) | 7.2 (0.46) | 3.32 (0.28) | 36.63 (2.02) |
| | Anchor (a=0.9) | 88.55 (0.14) | 7.53 (0.45) | 3.6 (0.23) | 37.78 (2.29) |
| | $\ell_2$ Reg (a=0.5) | 88.52 (0.19) | 7.86 (0.36) | 3.59 (0.18) | 40.38 (1.81) |
| | $\ell_1$ Reg (a=0.1) | 86.88 (0.35) | 8.24 (0.55) | 3.88 (0.41) | 36.81 (2.63) |
| | Co-distill (CE, a=0.5, $n_{warm}$=2k ) | 88.76 (0.21) | 7.51 (0.39) | 3.67 (0.3) | 37.98 (1.71) |
| | Co-distill (KL, 0.5, $n_{warm}$=2k) | 88.85 (0.35) | 7.83 (0.43) | 3.68 (0.29) | 40.59 (2.4) |
| | Bi-tempered ($t_1$=0.7, $t_2$=2) | 88.7 (0.29) | 7.36 (0.47) | 3.5 (0.19) | 37.24 (3.04) |
| | Mixup (a=0.4) | **89.17 (0.10)** | **6.77 (0.29)** | **3.23 (0.15)** | 35.97 (1.43) |
| | Control | 88.95 (0.26) | 9.13 (0.51) | 4.42 (0.4) | 46.99 (2.49) |
| CelebA Smiling | $k$-NN LS (k=100, b=0.1, a=0.9) | **90.02 (0.11)** | **5.46 (0.32)** | **2.97 (0.18)** | 27.71 (1.74) |
| | Label Smoothing (a=0.05) | 89.39 (0.29) | 6.77 (0.41) | 3.81 (0.26) | 31.67 (2.34) |
| | Anchor (a=0.8) | 89.87 (0.14) | 5.57 (0.28) | 3.07 (0.21) | 27.66 (1.38) |
| | $\ell_2$ Reg (a=0.01) | 89.35 (0.16) | 6.85 (0.34) | 3.92 (0.27) | 31.62 (1.21) |
| | $\ell_1$ Reg (a=0.5) | 89.39 (0.26) | 6.71 (0.26) | 3.61 (0.22) | 32.48 (1.35) |
| | Co-distill (CE, 0.5, $n_{warm}$=1k) | 89.59 (0.29) | 6.31 (0.23) | 3.66 (0.3) | 29.47 (1.47) |
| | Co-distill (KL, 0.5, $n_{warm}$=2k) | 89.57 (0.22) | 6.1 (0.23) | 3.34 (0.26) | 29.66 (1.47) |
| | Bi-tempered ($t_1$=0.9, $t_2$=2.) | 89.88 (0.18) | 6.44 (0.31) | 3.56 (0.19) | 31.96 (1.96) |
| | Mixup (a=0.2) | 89.71 (0.14) | 6.15 (0.12) | 3.51 (0.12) | 29.37 (0.66) |
| | Control | 89.67 (0.19) | 7.3 (0.45) | 4.06 (0.27) | 35.34 (2.35) |
| CelebA High Cheekbone | $k$-NN LS (k=100, b=0.1, a=0.9) | **84.48 (0.21)** | **7.7 (0.29)** | **4.64 (0.27)** | 24.44 (0.98) |
| | Label Smoothing (a=0.005) | 83.73 (0.17) | 8.68 (0.46) | 5.2 (0.35) | 26.61 (1.28) |
| | Anchor (a=0.9) | 84.48 (0.2) | 7.97 (0.39) | 4.77 (0.22) | 25.44 (1.58) |
| | $\ell_2$ Reg (a=0.001) | 83.6 (0.14) | 9.06 (0.32) | 5.41 (0.24) | 27.66 (1.03) |
| | $\ell_1$ Reg (a=0.01) | 83.59 (0.26) | 8.43 (0.23) | 4.93 (0.23) | 26.14 (1.16) |
| | Co-distill (CE, a=0.5, $n_{warm}$=1k) | 84.08 (0.21) | 8.96 (0.37) | 5.33 (0.36) | 28.11 (0.88) |
| | Co-distill (KL, a=0.5, $n_{warm}$=1k) | 84.31 (0.08) | 8.57 (0.16) | 5.06 (0.13) | 27.39 (0.47) |
| | Bi-tempered ($t_1$=0.5, $t_2$=4) | 83.92 (0.13) | 7.84 (0.32) | 4.75 (0.21) | 24.01 (1) |
| | Mixup (a=0.4) | 84.53 (0.14) | 7.92 (0.47) | 4.69 (0.31) | 25.53 (1.54) |
| | Control | 83.93 (0.56) | 10.18 (0.93) | 6.2 (0.89) | 31.1 (2.22) |
| Phishing | $k$-NN LS (k=500, a=0.8, b=0.9) | **96.69 (0.09)** | **1.04 (0.21)** | **0.54 (0.14)** | 15.81 (3.52) |
| | Label Smoothing (a=0.8) | 96.63 (0.09) | 1.26 (0.26) | 0.64 (0.17) | 18.8 (3.42) |
| | Anchor (a=0.9) | 96.02 (0.25) | 2.33 (0.25) | 1.08 (0.15) | 31.58 (5.11) |
| | $\ell_2$ Reg (a=0.5) | 96.51 (0.12) | 1.35 (0.3) | 0.7 (0.21) | 19.37 (4) |
| | $\ell_1$ Reg (a=0.5) | 95.38 (0.18) | 1.48 (0.34) | 0.83 (0.24) | 14.95 (4.08) |
| | Co-distill (CE, a=0.2, $n_{warm}$=2) | 96.02 (0.19) | 1.45 (0.26) | 0.83 (0.21) | 16.72 (4.13) |
| | Co-distill (KL, a=0.001, $n_{warm}$=1k) | 95.94 (0.33) | 1.51 (0.2) | 0.65 (0.18) | 20.95 (6.14) |
| | Bi-tempered ($t_1$=0.9, $t_2$=1.0) | 96.26 (0.37) | 2.32 (0.69) | 1.23 (0.53) | 30.19 (8.51) |
| | Mixup (a=0.1) | 96.22 (0.23) | 1.80 (0.33) | 1.05 (0.28) | 21.53 (4.25) |
| | Control | 96.3 (0.32) | 2.25 (0.59) | 1.21 (0.38) | 29.05 (7.93) |

Table 1: Results across all datasets and baselines under optimal hyperparameter tuning (settings shown). Note that we report the standard deviation of the runs instead of standard deviation of the mean (i.e. standard error) which is often reported instead. The former is higher than the latter by a factor of the square root of the number of trials (10).

### 5.3 Evaluation Metrics and Hyperparameter Tuning

For each dataset, baseline and hyper-parameter setting, we run each method on the same train and test split exactly 5 times. We then report the average test accuracy as well as the test set churn averaged across every possible pair $(i, j)$ of runs (10 total pairs). To give a more complete picture of the sources of churn, we also slice the churn by the whether or not the test predictions of the first run in the pair were correct. Then, lowering the churn on the correct predictions is desirable (i.e. if the base model is correct, we clearly don't want the predictions to be changing), while churn reduction on incorrect predictions is less relevant (i.e. if the base model was incorrect, then it may be better for there to be higher churn– however at the same time, some examples may be inherently difficult to classify or the label is such an outlier that we don't expect an optimal model to correctly classify in which case lower churn may be desirable). This is why in the results for Table 1, we bold the best performing baseline for churn on correct examples, but not for churn on incorrect examples.

In the results (Table 1), for each dataset and baseline, we chose the optimal hyperparameter setting by first sorting by accuracy and choosing the setting with the highest accuracy, and if there were multiple settings with very close to the top accuracy (defined as within less than $0.1\%$ difference in test accuracy), then we chose the setting with the lowest churn among those settings with accuracy close to the top accuracy. There is often no principled way to trade-off the two sometimes competing objectives of accuracy and churn (e.g. Cotter et al. (2019) offer a heuristic to trade off the two objectives in a more balanced manner on the Pareto frontier). However in this case, biasing towards higher accuracy is most realistic because in practice, when given a choice between two models, it's usually best to go with the more accurate model. Fortunately, we will see that accuracy and churn are not necessarily competing objectives and our proposed method usually gives the best result for both simultaneously.

### 5.4 Results

We see from Table 1 that mixup and our method, $k$-NN label smoothing, are consistently the most competitive; mixup outperforms on SVHN and Fashion MNIST while $k$-NN label smoothing outperforms on all the remaining datasets. Notably, both methods do well on accuracy and churn metrics simultaneously, suggesting that there is no inherent trade-off between predictive performance and churn reduction. Due to space constraints, ablations on SVHN for our method's hyperparameters $(a, b,$ and $k)$, along with results for the ensemble baseline can be found in the Appendix. While we found ensembling to be remarkably effective, it does come with higher cost (more trainable parameters and higher inference cost), and so we discourage a direct comparison with other methods.

## 6 Conclusion

Modern DNN training is a noisy process: randomization arising from stochastic minibatches, weight initialization, and data preprocessing techniques can lead to models with drastically different predictions on the same datapoints when using the same training procedure– and this phenomenon happens even when all the models attain similarly high accuracies.

Reducing such prediction churn is important in practical problems as production ML models are constantly updated and improved on. Since offline metrics usually can only serve as proxies to the live metrics, comparing the models in A/B tests and live experiments oftentimes must involve manual labeling of the disagreements between the models making it a costly procedure. Thus, controlling the amount of predictive churn can be crucial for more efficiently iterating and improving models in a production setting.

Despite the practical importance of this problem, there has been little work done in the literature on this topic. We provide one of the first comprehensive analyses of reducing predictive churn arising from retraining the model on the same dataset and model architecture. We show that numerous methods used for other goals such as learning with noisy labels and improving model calibration serve as reasonable baselines for lowering prediction churn. Moreover, we propose a new technique, $k$-NN label smoothing, which is shown to be a principled approach leveraging a local smoothing from the deep $k$-NN labels to enhance the global smoothing from the vanilla label smoothing procedure. We further show that it often outperforms the baselines across a range of datasets and model architectures.

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

## A    PROOFS

For the proofs, we make use of the following result from Jiang (2019) which bounds the number of distinct $k$-NN sets on the sample across all $k$:

**Lemma 1** (Lemma 3 of Jiang (2019)). *Let $M$ be the number of distinct $k$-NN sets over $\mathcal{X}$, that is, $M := |\{N_k(x) : x \in \mathcal{X}\}|$. Then $M \leq D \cdot n^D$.*

*Proof of Theorem 1.* We have by triangle inequality and the smoothness condition in Assumption 1 that:

$$|\eta_k(x) - \eta(x)| \leq \left| \sum_{i=1}^{n} (\eta(x_i) - \eta(x)) \cdot \frac{1\,[x_i \in N_k(x)]}{|N_k(x)|} \right| + \left| \sum_{i=1}^{n} (y_i - \eta(x_i)) \cdot \frac{1\,[x_i \in N_k(x)]}{|N_k(x)|} \right|$$

$$\leq C_\alpha \cdot r_k(x)^\alpha + \left| \sum_{i=1}^{n} (y_i - \eta(x_i)) \cdot \frac{1\,[x_i \in N_k(x)]}{|N_k(x)|} \right|.$$

We now bound each of the two terms separately.

To bound $r_k(x)$, let $r = \left( \frac{2k}{\omega \cdot v_D \cdot n \cdot p_{X,0}} \right)^{1/D}$. We have $\mathcal{P}(B(x,r)) \geq \omega \inf_{x' \in B(x,r) \cap \mathcal{X}} p_X(x') \cdot v_D r^D \geq \omega p_{X,0} v_D r^D = \frac{2k}{n}$, where $\mathcal{P}$ is the distribution function w.r.t. $p_X$. By Lemma 7 of Chaudhuri & Dasgupta (2010) and the condition on $k$, it follows that with probability $1 - \delta/2$, uniformly in $x \in \mathcal{X}$, $|B(x,r) \cap X| \geq k$, where $X$ is the sample of feature vectors. Hence, $r_k(x) < r$ for all $x \in \mathcal{X}$ uniformly with probability at least $1 - \delta/2$.

Define $\xi_i := y_i - \eta(x_i)$. Then, we have that $-1 \leq \xi_i \leq 1$ and thus by Hoeffding's inequality, we have that $A_x := \sum_{i=1}^{n} (y_i - \eta(x_i)) \cdot \frac{1[x_i \in N_k(x)]}{|N_k(x)|} = \sum_{i=1}^{n} \xi_i \cdot \frac{1[x_i \in N_k(x)]}{|N_k(x)|}$ satisfies $P(|A_x| > t/k) \leq 2 \exp\left(-t^2/2k\right)$. Then setting $t = \sqrt{2k} \cdot \sqrt{\log(4D/\delta) + D \log(n)}$ gives

$$\mathbb{P}\left( |A_x| \geq \sqrt{\frac{2\log(4D/\delta) + 2D \log(n)}{k}} \right) \leq \frac{\delta}{2D \cdot n^D}.$$

By Lemma 3 of Jiang (2019), the number of unique random variables $A_x$ across all $x \in \mathcal{X}$ is bounded by $D \cdot n^D$. Thus, by union bound,

$$\mathbb{P}\left( \sup_{x \in X} |A_x| \geq \sqrt{\frac{2\log(4D/\delta) + 2D \log(n)}{k}} \right) \leq \delta/2.$$

The result follows.                                                                                               $\square$

*Proof of Theorem 2.* Let $X$ be the $n$ sampled feature vectors and let $x \in \mathcal{X}$. Define $k'(x) := |X \cap B(x, r_\beta(x))|$. We have:

$$|\eta_k(x) - \widetilde{\eta}_\beta(x)| \leq |\eta_{k'(x)}(x) - \eta_k(x)| + |\eta_{k'(x)}(x) - \widetilde{\eta}_\beta(x)|.$$

We bound each of the two terms separately. We have

$$|k'(x) - k| = \left| \sum_{x \in X} 1[x \in B(x, r(x))] - \beta \cdot n \right|$$

By Hoeffding's inequality we have

$$\mathbb{P}(|k'(x) - k| \geq t \cdot n) \leq 2 \exp(-2t^2 n).$$

Choosing $t = \sqrt{\frac{\log(4D/\delta) + D \log(n)}{2n}}$ gives us

$$\mathbb{P}\left( |k'(x) - k| \geq \sqrt{\frac{n}{2} \cdot (\log(4D/\delta) + D \log(n))} \right) \leq \frac{\delta}{2D \cdot n^D}.$$

| Dataset (m=5) | Accuracy (%) | Churn (%) | Churn Correct | Churn Incorrect |
|---|---|---|---|---|
| SVHN | 90.34 (0.31) | 6.61 (0.19) | 2.75 (0.28) | 43.12 (1.49) |
| MNIST | 98.5 (0.07) | 0.94 (0.14) | 0.44 (0.09) | 33.74 (4.39) |
| Fashion MNIST | 89.71 (0.12) | 4.05 (0.14) | 1.85 (0.05) | 23.16 (1.29) |
| CelebA Smiling | 90.56 (0.09) | 3.35 (0.16) | 1.82 (0.11) | 17.95 (0.99) |
| CelebA High Cheekbone | 85.12 (0.16) | 4.95 (0.2) | 2.87 (0.1) | 16.81 (1.24) |
| Phishing | 96.11 (0.06) | 0.54 (0.08) | 0.29 (0.08) | 6.77 (1.31) |

Table 2: Ensemble results for all datasets. In all settings, the optimal $m$ (number of subnetworks) is 5. We see that compared to the other methods presented, ensembling does well in both predictive performance and in reducing churn. It does come at a cost, however: the model is effectively 5 times larger, making both training and inference more expensive.

By Lemma 3 of Jiang (2019), the number of unique sets of points consisting of balls intersected with the sample is bounded by $D \cdot n^D$ and thus by union bound, we have with probability at least $1 - \delta/2$:

$$\sup_{x \in \mathcal{X}} |k'(x) - k| \leq \sqrt{\frac{n}{2} \cdot (\log(4D/\delta) + D \log(n))}.$$

We now have

$$|\eta_{k'(x)}(x) - \eta_k(x)| \leq \left| \frac{1}{k} - \frac{1}{k'(x)} \right| \min\{k, k'(x)\} + \min\left\{ \frac{1}{k}, \frac{1}{k'(x)} \right\} |k - k'(x)|$$

$$\leq \frac{2}{k} \cdot |k - k'(x)| \leq \sqrt{\frac{2 \log(4D/\delta) + 2D \log(n)}{\beta \cdot n}}.$$

where the first inequality follows by comparing the difference contributed by the shared neighbors among the $k$-NN and $k'(x)$-NN (first term on RHS) and contributed by the neighbors that are not shared (second term on RHS).

For the second term, define $A_x := X \cap B(x, r_\beta(x))$. For any $x'$ sampled from $B(x, r_\beta(x))$, we have that the expected label is $\widetilde{\eta}_\beta(x)$. Since $\eta_{k'(x)}(x)$ is the mean label among datapoints in $A_x$, then we have by Hoeffding's inequality that

$$\mathbb{P}(|\eta_{k'}(x) - \widetilde{\eta}_\beta(x)| \geq k'(x) \cdot t) \leq 2 \exp\left(-t^2/2k'\right).$$

Then setting $t = \sqrt{2k'} \cdot \sqrt{\log(4D/\delta) + D \log(n)}$ gives

$$\mathbb{P}\left( |\eta_{k'(x)}(x) - \widetilde{\eta}_\beta(x)| \geq \sqrt{\frac{2 \log(4D/\delta) + 2D \log(n)}{k'(x)}} \right) \leq \frac{\delta}{2D \cdot n^D}.$$

By Lemma 3 of Jiang (2019), the number of unique sets $A_x$ across all $x \in \mathcal{X}$ is bounded by $D \cdot n^D$. Thus, by union bound, with probability at least $1 - \delta/2L$

$$|\eta_{k'(x)}(x) - \widetilde{\eta}_\beta(x)| \leq \sqrt{\frac{2 \log(4D/\delta) + 2D \log(n)}{k'(x)}}.$$

The result follows immediately for $n$ sufficiently large. $\qquad\square$

## B  ENSEMBLE RESULTS

In Table 2 we present the experimental results for the ensemble baseline. The method performs remarkably well, beating the proposed method and the other baselines on both accuracy and churn reduction across datasets. We do note, however, that ensembling does come at a cost which may prove prohibitive in many practical applications. Firstly, having $m$ times the number of trainable parameters, training time (if done sequentially) takes $m$ times as long, as does inference, since each subnetwork must be evaluated before aggregation.

| Fixed | Ablated | Accuracy (%) | Churn (%) | Churn Correct |
|---|---|---|---|---|
| k = 10, a = 1 | b = 0 | 86.54 (0.67) | 13.43 (0.58) | 5.86 (0.57) |
| | b = 0.05 | 87.37 (0.38) | 12.22 (0.31) | 5.34 (0.31) |
| | b = 0.1 | 86.94 (0.65) | 13.41 (0.39) | 5.69 (0.57) |
| | b = 0.5 | 88.48 (0.52) | 11.12 (0.5) | 4.37 (0.35) |
| | b = 0.9 | 88.98 (0.33) | 10.98 (0.28) | 4.64 (0.29) |
| k = 10, a = 0.5 | b = 0 | 84.44 (2.43) | 15.85 (2.39) | 6.73 (2.47) |
| | b = 0.05 | 79.64 (3.1) | 22.02 (5.15) | 10.28 (4.06) |
| | b = 0.1 | 79.88 (2.63) | 21.09 (3.59) | 10.25 (1.85) |
| | b = 0.5 | 84.44 (2.54) | 14.33 (1.78) | 6.52 (2.83) |
| | b = 0.9 | 81.06 (2.35) | 20.53 (4.52) | 8.68 (3.36) |
| k = 10, b = 0.9 | a = 0.005 | 73.91 (3.01) | 28.02 (5.66) | 13.85 (4.82) |
| | a = 0.01 | 72.41 (4.86) | 25.57 (5.78) | 13.66 (7.01) |
| | a = 0.02 | 72.03 (1.79) | 31.25 (7.25) | 17.26 (6.56) |
| | a = 0.05 | 73.2 (3.33) | 30.41 (6.2) | 17.96 (6.04) |
| | a = 0.1 | 75.28 (1.98) | 23.96 (4.76) | 10.13 (4.25) |
| | a = 0.5 | 81.06 (2.35) | 20.53 (4.52) | 8.68 (3.36) |
| | a = 0.8 | 85.99 (0.73) | 13.76 (0.75) | 6 (0.83) |
| | a = 0.9 | 87.27 (0.41) | 13.72 (0.41) | 5.68 (0.32) |
| | a = 1.0 | 88.98 (0.33) | 10.98 (0.28) | 4.64 (0.29) |
| k = 10, b = 0.5 | a = 0.005 | 71.45 (3.81) | 21.14 (4.37) | 11.5 (5.46) |
| | a = 0.01 | 74.73 (6.24) | 25.24 (3.84) | 8.28 (4.35) |
| | a = 0.02 | 73.59 (3.72) | 29.47 (6.89) | 17.52 (6.13) |
| | a = 0.05 | 74.17 (3.88) | 20.26 (4.15) | 5.79 (3.7) |
| | a = 0.1 | 72.43 (2.75) | 25.77 (5.41) | 13.42 (4.89) |
| | a = 0.5 | 84.44 (2.54) | 14.33 (1.78) | 6.52 (2.83) |
| | a = 0.8 | 87.26 (0.41) | 11.76 (0.24) | 4.62 (0.21) |
| | a = 0.9 | 86.85 (0.54) | 12.54 (0.44) | 5.25 (0.48) |
| | a = 1.0 | 88.48 (0.52) | 11.12 (0.5) | 4.37 (0.35) |
| a = 1, b = 0.9 | k = 10 | 88.98 (0.33) | 10.98 (0.28) | 4.64 (0.29) |
| | k = 100 | 88.19 (0.19) | 11.15 (0.23) | 4.67 (0.17) |
| | k = 500 | 87.98 (0.62) | 11.33 (0.35) | 4.72 (0.55) |

Table 3: Ablation on $k$-NN label smoothing's hyperparameters: $a$, $b$, and $k$ for the SVHN dataset.

## C  ABLATION STUDY

In Table 3, we report SVHN results ablating $k$-NN label smoothing's hyperparameters: $k$, $a$, and $b$. We observe the following trends: with $a$ fixed to 1, both accuracy and churn improve with increasing $b$, and a similar relationship holds as $a$ increases with $b$ fixed to 0.9. Lastly, both key metrics are stable with respect to $k$.

## D  HYPERPARAMETER SEARCH

Our experiments involved performing a grid search over hyperparameters. We detail the search ranges per method below.

**$k$-NN label smoothing.**

- $k \in [5, 10, 100, 500]$
- $a \in [0.005, 0.01, 0.2, 0.05, 0.1, 0.5, 0.8, 0.9, 1.0]$
- $b \in [0, 0.05, 0.1, 0.5, 0.9]$

**Anchor.**

- $a \in [0.005, 0.01, 0.02, 0.05, 0.1, 0.5, 0.8, 0.9, 1.0]$

$\ell_1, \ell_2$ **Regularization.**

- $a \in [0.001, 0.01, 0.05, 0.1, 0.2, 0.5]$

**Co-distill**

- $a \in [0.001, 0.01, 0.05, 0.1, 0.2, 0.5]$
- $n_{\text{warm}} \in [1000, 2000]$

**Bi-tempered**

- $t_1 \in [0.3, 0.5, 0.7, 0.9]$
- $t_2 \in [1., 2., 3., 4.]$
- $n_{\text{iters}}$ always set to 5.

**Mixup**

- $a \in [0.2, 0.3, 0.4, 0.5]$

**Ensemble**

- $m \in [3, 5]$

