# OpenReview forum: "Deep $k$-NN Label Smoothing Improves Reproducibility of Neural Network Predictions"
_ICLR.cc/2021/Conference — Reject_

### Official Review · AnonReviewer2 · 2020-10-27
**Good application with limited novelty, main objective is not well addressed**

**Rating:** 3
**Confidence:** 4

**Review:**

The main objective of this paper is to reduce the model stability, in particular, the prediction churn of neural networks. The prediction churn is defined as the changed prediction w.r.t. model randomness, e.g. multiple runs of networks. The paper proposed to use a interpolated version of global label smoothing and k-NN label smoothing. Theoretically it is shown that k-NN rule converges to the Bayes rule when k is small, and converges to a kernel smoothed version of Bayes rule when k is linear in n. Experiments are conducted that show the proposed method gives highest test accuracy and lowest churn rate in most cases.

The problem is interesting and the the paper is writen clearly. However, I think it is a little bit tricky that the main objective of this paper is not well addressed, and the originality seems limited.

1) Originality: The idea of using k-NN in label smoothing has been proposed in (Dara Bahri, Heinrich Jiang, and Maya Gupta. Deep k-nn for noisy labels. ICML, 2020), and I don't find much difference with the method in this paper (except for the interpolation with global smoothing). That paper, though does not has 'neural networks' in the title, also reported their experiments on deep learning models. Therefore, I think the submission is not very original in terms of methodology.

2) Theory: The main objective of the paper is to deal with model stability, i.e. the churn rate. Thus, I would expect some results particularly regarding that aspect. However, the theorem 1 and 2 both only state the relationship between the k-NN rule and Bayes rule. What's the relatioinship between these results and the churn rate? If such results (at least any insight) can be discussed more, the paper would be better. The current theorems, to some extend, are like independent results on k-NN theory, stating that k-NN rule converges to the Bayes rule, which is already known in literature (perhaps under different assumptions). Taking a closer look I found that theorem 1 is actually pretty similar to the results provided in (Deep k-nn for noisy labels, ICML'20). I suggest the authors to highlight the difference and signifiacnce of the theorems, compared to many existing convergence restuls in literature. Also, the results are all for Definition 2, while Definition 3 is actually used. What can we say about Definition 3?

3) Experiments: I suggest also provide loss and accuracy curves in figures, instead of only the test accuracy at the end. I'm aware that two more parameters, a and b, are fine tuned in the proposed method. Since only the best result is reported, I'm wondering how does it perform when a and b is not tuned very carefully? For example, it might be convenient to set some default a and b value. It would be helpful to see some results (on different a and b) in the appendix.

Overall, in my understanding the key of this submission is to argue that k-NN label smoothing helps with model stability, on neural networks (otherwise the originality would be very poor). However, the discussion and results on this respect seems insufficient. Though the empirical performance is OK, I think the paper needs to improve the theoretical results and provide more insight and implication on model stability.

===================================================

I have read the author response, and I have to downgrade my score to 3 because of the validity of experiments. A minor issue is that the theoretical result is not impressive since the technique is basically the same as ICML'20 paper, except that expectation is taken here which makes things easier.

My major concern is about the experimental results. The additionally provided Table 3 makes me question the implementation of the experiments. For example, a=0.005 should be very close to the true label (by Definition 3), but the accuracy is only around 70%. For LeNet on SVHN dataset, the accuracy of training with true label y should approach 90%, e.g. see the report in [Coverage Testing of Deep Learning Models using DatasetCharacterization, 2019]. Indeed, currently best model would give <5% test error. Such a big gap is questionable. The only reason I can think of is that this is a typo and larger 'a' should actually mean more weight on y. Then, the results say that a=1 (corresponds to the true label without smoothing) gives best performance (around 90% which makes sense). However, this actually means that the proposed method is ineffective. In any case, for me the validity of experiments is poor.

I'm fairly confident in my evaluation. However, if any reviewer or area chair points out that I made a mistake, please kindly correct me and I will re-evaluate the work. Currently, since the experimental results are questionable for me, I would suggest a rejection of this paper.

---

> ### Author Response · Authors · 2020-11-25
> **Response to Review**
>
> Originality: the method in Bahri et al (2020) does not involve any sort of  label smoothing and  solves a different problem of filtering examples that are likely wrongly labeled via disagreement of the hard label and the labels of the k-nearest neighbors. Here, we modify the labels into a soft one by averaging with the k-NN label (as well as uniform distribution) to solve the predictive churn problem. Thus, the methods are also quite different.
>
> Theory: Theorem 1 in Bahri et al (2020) (as well as their other results) are quite different from the results we provide for the following reasons:
> Bahri et al (2020) is concerned with recovering the Bayes-optimal *hard* label. That is \eta*(x) = 1[\eta(x) >= 0.5] while we are concerned with recovering the Bayes-optimal *soft* label. That is, \eta*(x) = E[\eta(x)]. (This is because Bahri et al (2020) is a hard-label method while ours is a soft label smoothing method).
> The point of the theory in Bahri et al (2020) is to show that kNN disagreement can identify suspiciously labeled examples in the presence of corrupted examples: their results show rates in terms of the “minimum pairwise distance” of corrupted examples. Here, we don’t have any corrupted data points and assume all examples are i.i.d..
> Bahri et al (2020) assumes a Tsybakov margin condition, while we do not make (or need) such an assumption. Thus, while some elements of the analysis may be similar, they are still very different proofs.
> We provide the first convergence rates for any sort of kNN procedure in the setting where k grows linearly with n. All other convergence results in the literature require k / n -> 0 for convergence.
>
> The effect on the churn rate is quite difficult to quantify because this would depend on the training procedure used as well as model architecture. Our results show that the kNN label smoothing technique may have similar effects as using distillation-- however there is also very little known about distillation theoretically despite being a very popular method.
>
> Following your suggestion, we've updated the paper's appendix to include an ablation of our method's hyperparameters ($k$, $a$, and $b$) on the SVHN dataset.

---

### Official Review · AnonReviewer1 · 2020-10-29
**Pragmatic solution to a pressing problem in practical machine learning with non-trivial theory to back it up**

**Rating:** 7
**Confidence:** 2

**Review:**

My enthusiasm for this paper remains fairly high after reading through the other reviews and responses: I think the results (in the main body of the paper) look strong. In particular, this approach is competitive with MixUp, and I prefer this simpler label-only approach to MixUp. I don't think this work is duplicative with the ICML 2020 deep knn paper cited by another reviewer (that paper filters out noisy examples, rather than smoothing labels). And I think the theory is novel in its synthesis oof KNN theory with label noise.

However, I am unnerved by something that Rev 2 pointed out about the ablation results in the new Table 3 in the appendix: in Definition 3, the weight given to the assigned ("true") label is $(1-a)$, so we'd expect to see higher (or at least competitive) accuracy for lower values of $a$ when label noise is not too severe. For SVHN, at least, this seems like it should be true since we know the SOTA accuracy is in the 90s.

Instead we see the highest accuracy is for the highest values of $a$. That makes me suspect the order is reversed. However, if this method works, we should expect to see lower churn for higher values of $a$, which we do, which would indicate that things are in the right order. This apparent contradiction is disconcerting since some of these numbers are cited in the comparison in Table 1 in the main paper.

In an ideal world, it'd be nice to have one more round of discussion with the authors about Table 3. Because it was added to the appendix late in the review process and no further discussion is possible, I am electing to leave my score as is. However, I urge the authors to take a look at Table 3 and verify that there is no error. If it is possible for the authors to post a public comment and/or contact the AC with an explanation, they should do so.

I'm also lowering my confidence to reflect my uncertainty about Table 3.

-----

This submission studies the problem of "prediction churn," in which independently trained models possessing equivalent accuracy produce different predictions for individual (usually novel, sometimes out-of-sample) samples. It proposes a solution built around a "local" K-nearest neighbor (KNN) variant of "label smoothing" and demonstrates in fairly thorough experiments that it outperforms (roughly) half a dozen alternatives. The intuition for its success is that KNN label smoothing reduces label noise (but with greater precision than global smoothing). The paper backs up this argument with theory from KNN, providing a uniform bound between the KNN smoothed label and the Bayes optimal label.

I enjoyed this paper, and I tentatively support its inclusion in the conference. The manuscript conveniently summarizes its claims in the introduction:
1. A (first of its kind?) thorough empirical comparison of (roughly) half a dozen approaches (plus some variants) to reducing prediction churn, including several that were introduced in other contexts, as well as the proposed approach.
2. A proposed solution to churn based on a local (KNN) variant of label smoothing that consistently outperforms the included baselines in the described experiments.
3. A non-trivial theoretical analysis that supports intuitive explanation for the success of KNN label smoothing.

In my view, these claims are justified, though I should add some caveats. I cannot fully substantiate (1), but a cursory skim of the referenced literature and a brief Google Scholar search suggest that this is indeed the first such direct empirical comparison of the included baselines specifically for the problem of churn. For (2), the proposed method is clever and well-motivated and appears effective in practice. For (3), I was not previously familiar with the recent theory on KNN models (especially Jiang 2019), and I did not rigorously check either Theorem or the proofs in the appendix.

The introduction does a nice job of motivating the problem of prediction churn, which I suspect will resonate with practitioners. The text argues (correctly, in my opinion) that this problem can be highly vexing -- and costly -- in production machine learning settings but has received proportionally little attention in the academic literature.

The proposed method is a non-trivial extension of an existing method (global label smoothing) that was simpler and designed to address calibration, rather than churn. The extension is intuitive but does not immediately follow from previous work: mix hard labels with a more informative distribution based on each data point's neighborhood of samples, rather than a uniform distribution. The discussion and theoretical analysis provide context for previous work and deepens understanding of, in particular, label smoothing.

The experiments are well-designed and thorough: they include five different datasets and do a rigorous job of tuning each model sufficiently to ensure a fair comparison. The proposed churn "metrics" are well-motivated and sensible. I especially like the separate quantifications of correct and incorrect churn rates. The proposed method consistently outperforms the included alternatives, though admittedly not always by a substantive margin. As the paper observes, it was heartwarming to see that for at least the studied datasets, there is not always a meaningful tradeoff between accuracy and churn.

The theoretical analysis is very interesting, especially its connection to existing theory for KNN models (and my non-expert interpretation is that this research in fact builds on previous results). I especially appreciated the last two paragraphs of Section 4.1, which provide an intuitive interpretation of the theory in terms of reducing label noise in a way that should be optimal in the limit under non-onerous conditions.

My enthusiasm for the submission is somewhat diminished by a handful of flaws (none fatal), mostly to do with clarity of exposition or missing details. These flaws should be addressable by the authors.

For one, it is not entirely clear from the body of the paper in which space the nearest neighbors are computed: the original feature space, an intermediate representation from a hidden layer of a neural net, etc. Line 4 of Algorithm 1 suggests the neighbors are computed using the logits $z_i$, but this is not reinforced in the prose, nor in the theoretical analysis. It's a crucial implementation detail and should be stated clearly. If there is a choice, then the pros and cons of each should be discussed explicitly.

On a that note, if there is a choice (inputs, hidden layers, logits), then this suggests an additional experiment comparing results for each.

The theory is dense, particularly for readers not previously familiar with the KNN literature. The prose at the end of Section 4.1 helps, but there is a still a gap between the bounds shown in the Theorems and the prose explanation. I would suggest adding some additional language that explicitly connects the intuitive conclusion to the bounds themselves.

Finally, the fact that 4/5 datasets are images leaves me to wonder whether and how these results generalize to other domains. This becomes even more important if the label smoothing is based on nearest neighbors in the original input space: designing a distance for, e.g., a mix of continuous and categorical features is tricky. According to the UCI repository, the Phishing dataset is relatively low dimensional (30 features) and homogeneous. I would suggest adding at least one dataset with higher dimensional, sparse or heterogeneous input spaces, which are common in the production machine learning settings that the manuscript uses to motivate this work.

My initial recommendation is to accept this submission, but I reserve the right to revise my rating up or down, especially if the other reviewers discover key flaws that I missed. I urge the authors to seize the opportunity to improve the manuscript.

I have a few lingering questions:
- How sensitive is this method to choice of K?
- How sensitive is this method to the choice of similarity/distance or "quality" of the nearest neighbors? Suppose you start injecting spurious (or "adversarial") neighbors?
- Since the text connects label smoothing to label noise, how robust is this method to (a) higher rates of noise and (b) non-uniform label noise, e.g., class conditional or even feature-dependent noise?

---

> ### Author Response · Authors · 2020-11-25
> **Response to Review**
>
> We thank you for the detailed review.
>
> We have updated the paper with the following changes, to address your comments:
> 1) We have clarified that the $k$-NN is computed in "logits" space in the main text, instead of only in Algorithm 1.
> 2) To understand the sensitivity on $k$, we have added an ablation study on $k$ in the appendix. We see that both predictive performance and churn reduction is mostly stable with respect to $k$. We also ablate the method's two other hyperparameters, $a$ and $b$.

---

### Official Review · AnonReviewer3 · 2020-10-29
**The contribution significance may not be enough. The effectiveness of the proposed method for general problems may be limited.**

**Rating:** 5
**Confidence:** 3

**Review:**

This paper condsidered the problem of churn reduction in DNN training,  and proposed to  utilize the k-NN predictions to smooth the labels results. Theorectical  analysis  and empirical  comparison results were given to validate the proposed  idea.

Overall, the paper is well written. Main ideas and key contributions are clearly conveyed.  In my opinion, the studied churn reduction problem is of considerable interest in both academical and practical aspects.

My main concerns are about the contribution significance and practical effectiveness of the proposed method for general problems. While I'm not an expert on theoretical knn analysis, from my understanding, the thoeretical analysis seems straightforward extension/application of previous work, which may limit the technical contribution of this paper.

Secondly,  for the empirical validation, it's not clearly explained how the experiment setting, e.g., the model architechture, the train epochs,  the hyperparamter selection criteria, are designed, which hurts the reliability of the experiment results.

---

> ### Author Response · Authors · 2020-11-25
> **Response to Review**
>
> We thank the reviewer for appreciating the importance of the churn reduction problem.
>
> We believe the theoretical analysis is different from previous work: we provide the first finite-sample uniform convergence rates for k-NN classification (previous works use weaker notions of statistical consistency and/or deal with k-NN regression) and we also provide the first analysis which shows what happens when k grows linearly with n (previous works require k / n -> 0 in order to obtain convergence).
>
> For the empirical validation, we precisely explained the model architectures as well as the number of training epochs used for each dataset in Section 5.2 (see the bullet points); the hyperparameter selection criteria is described in the last paragraph of Section 5.3.

---

### Official Review · AnonReviewer5 · 2020-11-04
**Review: DEEP k-NN LABEL SMOOTHING IMPROVES STABILITY OF NEURAL NETWORK PREDICTIONS**

**Rating:** 5
**Confidence:** 4

**Review:**

##########################################################################
Summary:

DNNs from multiple runs each of which follows the same training procedure produces different predictions that lead to prediction churn despite the higher accuracies of the deep models. The paper proposes a k-NN label smoothing approach in order to reduce the churn as opposed to the traditional label smoothing.

##########################################################################

Reasons for score:

Overall, I vote for below the acceptance threshold in its current form. The idea of improving the prediction churn is encouraging. The k-NN label smoothing with theoretical guarantees for the bounds sounds interesting. However, the paper has serious drawbacks in the experimental evidence. The paper requires some proofreading. The state-of-the-art on label smoothing is not upto date. My concerns are further detailed in the comments below. Hopefully the authors will address these concerns in the rebuttal period.
##########################################################################


Pros:
Overall, the paper is written reasonably fine with a few grammatical errors, easy to follow and understandable.

The motivation for the paper comes from the inherent noise in random initialization, batch ordering, data augmentations and/or other preprocessing tricks.

The L_{\infinity} bounds with uniform guarantee is a positive compared to the average case in the literature.
k-NN label smoothing is interesting

Cons:
“... even when the trained models all attain high accuracies” does not flow well, correct it.

“... labels results in a new and principled ...” contains a mistake in plural and/or tense, please correct

Not clear why preprocessing is inherently noisy. Better point out to a specific preprocessing or a class of those kinds of techniques. Because, for example the normalization of data to 0 mean and 1 standard deviation is proved to smoothen the optimization plain and is more useful rather than being so bad on the loss surface.
“... train a model the model on the soft labels ...” does not flow well, correct it.

“... rather than a the pure global ...”, needs a fix.

One of the reasons from the motivation behind the churn is data augmentations. However, none of the augmentations are experimented in the results section.

In fact, the recent augmentation, MixUp is supposed to perform better in terms of calibration or improving the confidence in predictions.

Zhang, H., Cisse, M., Dauphin, Y.N. and Lopez-Paz, D., 2018, February. mixup: Beyond Empirical Risk Minimization. In International Conference on Learning Representations.

“Guo, C., Pleiss, G., Sun, Y. and Weinberger, K.Q., 2017, July. On Calibration of Modern Neural Networks. In International Conference on Machine Learning (pp. 1321-1330).”

“Sunil Thulasidasan, Tanmoy Bhattacharya, Jeff A. Bilmes, Gopinath Chennupati, Jamal Mohd-Yusof: Combating Label Noise in Deep Learning using Abstention. ICML 2019: 6234-6243.”

Thulasidasan et al, showed that MixUp returns well calibrated confidence scores compared to label smoothing. In which case, it might be worth comparing against the state-of-the-art rather simply the label smoothing. Note that Mixup has a kind of label smoothing effect.

More recent techniques such as AugMix and Deep Ensembles are said to be better in producing highly confident predictions.

Hendrycks, D., Mu, N., Cubuk, E.D., Zoph, B., Gilmer, J. and Lakshminarayanan, B., 2019. Augmix: A simple data processing method to improve robustness and uncertainty. arXiv preprint arXiv:1912.02781.

“Lakshminarayanan, B., Pritzel, A. and Blundell, C., 2017. Simple and scalable predictive uncertainty estimation using deep ensembles. In Advances in neural information processing systems (pp. 6402-6413).”

Fort, S., Hu, H. and Lakshminarayanan, B., 2019. Deep ensembles: A loss landscape perspective. arXiv preprint arXiv:1912.02757.

The question is does the churn happen even with the ensemble approaches, which is supposed to produce stable predictions? It will be interesting, if the motivation behind this paper is to show that these more advanced models still suffer from the churn problem and then show the reductions, etc.

The benchmarks are small, may be showing the method working on ImageNet scale data will strengthen the paper.

---

> ### Author Response · Authors · 2020-11-25
> **Response to Review**
>
> We thank the reviewer for appreciating the idea of improving churn reduction as well as finding the theoretical results interesting.
>
> It seems that the reviewer’s main concern is lack of baseline results for data augmentation methods (i.e. MixUp) and ensemble methods. We thank the reviewer for bringing these up as possible baselines for churn.
>
> We have uploaded an updated version of the paper which includes experimental results for MixUp and ensemble.
>
> MixUp: We found that MixUp was competitive to our method. MixUp however requires us to augment both the features and the labels while our method only modifies the labels. Moreover, MixUp would not be an appropriate choice for datasets and models with categorical features as the convex combinations between them may be undefined.
>
> Ensemble: We found the ensemble method to generally outperform all of the baselines; however, we stress that the ensemble method leverages a more complex model class (since the number of parameters in the final model is $m$ times as high as that of the other baselines where $m$ is the number of constituents in the ensemble). Thus, the comparison is not completely fair since the ensemble method’s model architecture will have more learning power than that of the other baselines. Moreover, in latency sensitive production settings, it may be quite undesirable to perform inference on an ensemble.
> Still, this shows that the ensemble approach is a powerful one which we will highlight in the paper.
>
> We again thank the reviewer for bringing up these baselines to try which ended up being powerful alternatives to our kNN label smoothing method. We believe these results made the paper provide further insights into solving the churn problem-- an important practical problem that has thus far received little treatment in the literature.

---

### Author Response · Authors · 2020-11-25
**General Response**

We thank the reviewers for the thoughtful and detailed reviews.

We updated the paper based on some of the suggestions. Please take a look.

---

### Decision · Program_Chairs · 2021-01-07
**Final Decision**

**Decision:**

Reject

**Comment:**

This paper proposes a k-NN smoothing procedure for dealing with the problem of churn prediction. The idea is interesting and is based on theoretical foundations. The reviews have raised some limitations in the significance and in the experiments. The rebuttal provided by the authors have addressed some concerns. However, the new experimental evaluation have raised new concerns about the results, in particular with respect to results given in Table 3. Some typo may exist, but even some doubts remain on the experimental evaluation and results and thus on the effectiveness of the results.
Authors' rebuttal was too late to allow another round of discussion.
Considering the current concerns and uncertainties on the paper, I have to recommend rejection.